# Low Intracellular Water, Overhydration, and Mortality in Hemodialysis Patients

**DOI:** 10.3390/jcm9113616

**Published:** 2020-11-10

**Authors:** Carolina Gracia-Iguacel, Emilio González-Parra, Ignacio Mahillo, Alberto Ortiz

**Affiliations:** 1Renal Medicine, IIS-Fundación Jiménez Díaz UAM University Hospital, 28040 Madrid, Spain; EGParra@quironsalud.es (E.G.-P.); Imahillo@fjd.es (I.M.); aortiz@fjd.es (A.O.); 2Instituto Reina Sofia de Investigaciones Nefrologicas (IRSIN-FRIAT), 28003 Madrid, Spain; 3Medicine Department, Universidad Autonoma de Madrid, 28049 Madrid, Spain

**Keywords:** hemodialysis, mortality, outcomes, nutrition, overhydration, bioimpedance

## Abstract

Background: In hemodialysis patients, extracellular water (ECW) overload predicts all-cause and cardiovascular mortality. The primary aim of the present study was to analyze changes in post-dialysis (i.e., following removal of excess ECW) ECW, intracellular water (ICW), and the overhydration (OH) parameter over time. Additionally, the association of these parameters with mortality was explored. Patients and methods: Prospective study of prevalent hemodialysis patients (*n* = 124) followed for a median of 20 (interquartile range (IQR) 8–31) months. In three visits, inflammation (C-reactive protein) and post-dialysis fluid status (bioimpedance, BIS) were assessed. Results: During follow-up, the overhydration (OH) parameter increased (−0.696 ± 1.6 vs. 0.268 ± 1.7 L; *p* = 0.007) at the expense of a decrease in intracellular water (ICW) (19.90 ± 4.5 vs. 18.72 ± 4.1 24 L; *p* = 0.006) with a non-significant numerical increase in ECW/ICW ratio (0.795 ± 0.129 vs. 0.850 ± 0.143; *p* = 0.055). Baseline ICW positively correlated with muscle mass and energy intake and negatively with C-reactive protein and it was lower in those who died than in survivors (15.09 ± 2.36 vs. 18.87 ± 4.52 L; *p* = 0.004). In Kaplan–Meier analysis, patients with low baseline ICW (≤17 L) and high ECW/ICW ratio (≥0.84) were at an increased risk of death. Baseline ICW was also associated with the risk of death in adjusted Cox proportional hazards models (HR 0.62 (0.40–0.98) *p* = 0.04). Conclusions: In hemodialysis patients, the post-dialysis OH parameter increased over time while ICW decreased, without changes in ECW. Low baseline post-dialysis ICW correlated with muscle wasting and inflammation and was an independent risk factor for mortality.

## 1. Introduction

The prime function of the kidney is to maintain a stable milieu interieur by the selective retention or elimination of water, electrolytes, and other solutes. Fluid overload has been independently related to cardiovascular and all-mortality in hemodialysis patients [1,2,3]. Despite new dialysis techniques allowing increased fluid removal, target weight may not always be reached due to increased intradialytic symptoms and fluid overload still remains a mortality risk factor. As dialysis patients present changes in both volume and distribution of fluid between different compartments, the fluid excess may be difficult to estimate and to remove [4].

Most analyses showing associations between fluid overload and mortality have focused on the excess extracellular water (ECW) assessed by bioelectrical Impedance Spectroscopy (BIS) prior to the dialysis session [1,2,3,5]. Assessment of fluid status is key to dialysis prescription, as pre-dialysis fluid overload is related to hypertension, left ventricular hypertrophy, high pulse wave velocity (PVW), N-terminal pro-brain natriuretic peptide (pro-BNP), and heart failure [6,7,8,9] while fluid depletion may result in intradialytic hypotension, tissue ischemia, and loss of residual renal function [2]. Pre-dialysis fluid overload was thought to result from increased ECW and to be associated with inflammation [10] and malnutrition [11] without being able to clarify whether increased ECW was a cause or consequence of those. The recent KDOQI clinical practice guideline for nutrition in CKD 2020 suggests the use of multifrequency BIS to assess body composition. FM (fat mass) and FFM (fat free mass) measured using multifrequency BIS had good agreement with DXA and had high correlations with several markers of nutritional status.

We hypothesized that despite removing excess fluid during the hemodialysis session, dialysis patients have changes in body composition, likely related to inflammation and muscle wasting, that lead to “subclinical” overhydration driven by decreased ICW or to a pathological post-dialysis ECW/ICW ratio driven by low ICW. The primary aim of the present study was to analyze changes in post-dialysis ECW, intracellular water (ICW), and the overhydration (OH) parameter over time. Additionally, the association of these variables with mortality was explored. This study highlights the use of BIS in dialysis patients to quantify the ICW compartment and potentially correct subclinical overhydration by increasing energy intake and avoid muscle wasting, for instance, by prescribing an exercise program.

## 2. Patients and Methods

### 2.1. Patients

This is a secondary analysis of a prior cross-sectional study with prospective follow-up of prevalent dialysis patients conducted at the Iñigo Alvarez de Toledo Kidney Foundation (FRIAT) hemodialysis center of the Fundación Jiménez Díaz Hospital, Madrid, Spain [12]. The primary aim was to analyze changes in post-dialysis (i.e., following removal of excess ECW) ECW, intracellular water (ICW), and the overhydration (OH) parameter over time. For this, no formal sample size was calculated. Next, we explored the variables associated with these changes. Finally, we explored the association of these variables with mortality. All prevalent patients (*n* = 124) on maintenance hemodialysis at the dialysis center were screened and the study ran from 1 January 2010 and survival was calculated until 1 October 2012 with a median follow-up of 461 days (240–931 days). The study consisted of three visits in which clinical, biochemical, anthropometric, and body composition parameters were collected along with dialysis characteristics and the presence of Protein Energy Wasting (PEW) was evaluated annually (baseline, 12 and 24 months). The cohort has been described in more detail elsewhere [12]. Patients were clinically stable. The study was approved by the local Ethical Committee and signed informed consent was obtained from all included patients. The ethical approval code: EO133-20_FJD.

Dialysis therapy was performed for at least 4 h thrice weekly, using ultrapure water. The dialysate calcium concentration was 1.25 or 1.50 mmol/liter. Adequacy of dialysis was estimated by mid-week single-pool Kt/V_urea_ using pre- and post-dialysis blood urea nitrogen (BUN) levels and pre- and post-dialysis body weight [13]. Mean KTV_urea_ was 1.42 (1.24 to 1.6). Membranes were high-flux polysulfone (27%), low-flux polysulfone (41%), and high-flux polynephron (32%). No patient was lost to follow-up.

### 2.2. Biochemistry

Blood samples were drawn under fasting conditions before the dialysis session between 8:00 a.m. and 12:00 p.m., after 20–30 min of quiet resting in a semi-recumbent position. Serum creatinine, calcium (Ca), phosphate (P), albumin, and C reactive protein (CRP) were analyzed using certified methods at the Biochemistry Laboratory at Fundación Jiménez Díaz.

### 2.3. Body Composition, Fluid Volume Parameters, and Nutritional Status

Anthropometric and body composition measurements were performed immediately after the mid-week dialysis session by a single observer on the same day that blood samples were collected.

The body mass index (BMI) was expressed in kg/m^2^. Weight was calculated as dry weight, defined as post-dialysis weight in which the patient was normotensive and with no signs of fluid overload. Body surface area (BSA) was estimated in cm^2^ by the equation: BSA = 0.0003207 × (weight)^0.7285−0.0188xlog(weight)^ × (height)^0.3^. The triceps fold (TF) was measured in millimeters with a plicometer in triplicate (lipocalibre Holtain, Crymych, United Kingdom) in the arm contralateral to the vascular access. The brachial circumference (BC) was measured in centimeters in the middle third with a flexible tape measure (Holtain Ltd., Crymych, United Kingdom). TF and BC were used to calculate the mid-upper arm muscle circumference (MUAMC) using the formula MUAMC = BC − (0.314 × TF) [14]. Body composition analysis was performed post-dialysis, after a 30-min rest period, every six months by Bioelectric Impedance Analysis (BIS) using the Fresenius Medical Care BCM body composition monitor, multi-frequency Bio-impedance analysis technology. Electrodes were placed in the contralateral side to the vascular access location. Three compartments (LTM (lean tissue, primarily muscle), ATM (adipose tissue), and OH (overhydration)) were identified from weight, height, intracellular water (ICW), and extracellular water (ECW) measurements [15]. Overhydration is calculated from the difference between the measured ECW and the predicted values based on fixed hydration on lean and adipose tissue mass. To avoid inter-observer variation, a single well-trained nephrologist (CG-I) performed all BIS. The normalized protein catabolic rate (nPCR), expressed as g/kg per day, was estimated from the Kt/V_urea_, an index of urea removal during dialysis, and average BUN (midweek) as follows: nPCR = (0.0136 × (Kt/V_urea_ × ((predialysis BUN + postdialysis BUN) − 2))) + 0.251 (13). In addition, blood pressure, pulse pressure, rate of dialysis ultrafiltration, interdialysis body weight gain, and symptoms associated with hypotension were also recorded. Malnutrition was defined according to the classification of PEW (ISRNM 2008). The diagnosis of malnutrition, as defined by PEW, requires the joint evaluation of the combination of clinical, anthropometric, and body composition parameters using bioelectrical impedance (BIA multifrequency) and biochemical parameters. PEW was assigned to patients meeting at least three criteria in the four different categories for malnutrition markers at baseline: biochemical (albumin < 3.8 g/dL, prealbumin < 30 g/dL or cholesterol < 100 mg/dL); body mass (BMI< 23 kg/m^2^, weight loss > 5% in three months or > 10% in six months or body fat < 10%); muscle (muscle mass loss > 5% in three months or > 10% in six months or MUAMC reduction ≥ 10% associated with the 50th percentile in the population); Protein intake (nPCR < 0.8 g/kg/day) [16]. A history of comorbidities was recorded for each patient and scored according to Charlson et al. [17].

The presence of fluid overload was defined as OH parameter > 1.1 L [18] and high ECW/ICW (>0.7). At 12 months (visit 1) and at 24 months (visit 2), fluid status and changes in the three compartments were assessed by BIS and inflammation by CRP.

### 2.4. Statistical Analyses

Statistical analyses were performed using R (version 3.0.1). Normally distributed variables are expressed as mean ± SD and non-normally distributed variables as median and range (minimum and maximum) or interquartile range (25th–75th percentile, IQR). Categorical values are expressed as number and percentage. To analyze changes in body composition, the same variable at different visits, we performed a Wilcoxon test for related samples. Two groups were compared using the Mann–Whitney or χ^2^ tests or the Fisher’s Exact test when frequencies of occurrence were less than five. Differences between more than two groups were analyzed by the Kruskal–Wallis test, as many values were not normally distributed. Spearman’s rank correlation (ρ) was used to determine the univariate correlation between ICW levels and selected parameters. Multivariate associations were performed by multivariable regression analyses when studying the determinants of low ICW values. To analyze mortality, survival curves were estimated using the Kaplan–Meier method and the log-rank test was used for comparisons. Univariable Cox regression models for mortality were estimated, and a multivariable Cox regression model was constructed, taking as potential predictors those variables with a *p*-value less than 0.2 in the univariable models. For maximum discrimination of survival by OH, low ICW and ECW/ICW ratio cut off values were calculated by the maximally selected rank statistics method, available in the R maxstat package. All comparisons were performed with bilateral tests and a 0.05 level of significance.

## 3. Results

### 3.1. Patient Characteristics

Hemodialysis patients were clinically stable; their mean age was 64 ± 13 years, 49 (45%) were women and dialysis vintage was 59 ± 79 months. The causes of ESRD were chronic glomerulonephritis (*n* = 22; 20%), interstitial nephritis (*n* = 12; 11%), polycystic kidney disease (*n* = 12; 11%), vascular (*n* = 31; 28.4%), diabetic nephropathy (*n* = 16; 15%), unknown (*n* = 13; 12%), and other causes (*n* = 3; 2.8%). A total of 23 patients (21%) were diabetic and 44 (40%) presented a cardiovascular disease (CVD). Vascular access was arteriovenous fistula (*n* = 100; 92%) or permanent catheter (*n* = 9; 8%). The characteristics of the cohort at the time of inclusion are shown in Table 1.

Malnutrition defined as PEW was present in 44 of 108 (41%) patients. Of the total population, 49% had an albumin level < 3.8 g/dL, 35% had prealbumin levels < 30 mg/dL and only 2.7% had cholesterol levels < 100 mg/dL. Among the body composition and anthropometric parameters, 41.6% had a BMI < 23 kg/m^2^, 42% had a MUAMC below the 50th percentile and 73% had an insufficient protein intake, defined as nPCR < 0.8 g/kg/day.

### 3.2. Baseline Post-Dialysis Body Composition Parameters

Baseline post-dialysis body composition parameters are described in Table 1.

**Table 1 jcm-09-03616-t001:** Baseline characteristics of the 124 dialysis patients.

Variables	Mean ± SD% (*n*) or Median (IQR)	Variables	Mean ± SD% (*n*) or Median (IQR)
Age (years)	64 ± 13	Serum C Reactive Protein (mg/dL)	1.3 ± 0.6
Women (% *n*)	45 (49)	Serum albumin (g/dL)	3.8 ± 0.4
Diabetes (% *n*)	21 (23)	nPCR (g/kg/day)	0.7 ± 0.3
Vintage (months)	59 ± 79	Plasma BNP (pg/mL)	6090 (2407–14900)
Cardiovascular disease (% *n*)	40 (44)	Plasma total CO2 (mmol/L)	20 ± 3
Predialytic systolic blood pressure	132 ± 24	Total Body Water (L)	32.8 ± 6.8
Predialytic diastolic blood pressure	67 ± 14	Extracellular Water (ECW) (L)	14.4 ± 2.6
Protein Energy Wasting (% *n*)	44 (41%)	Intracellular Water (ICW) (L)	18.5 ± 4.5
kTv	1.42 ± 0.18	ECW/ICW	0.79 ± 0.13
Hemoglobin (g/dL)	11.6 ± 1.4	Lean Tissue Mass (kg)	58 ± 16
Plasma iPTH (pg/mL)	250 (127–454)	Adipose Tissue Mass (kg)	30 ± 11
Serum calcium (mg/dL)	8.8 ± 0.8	Overhydration (L)	−0.70 ± 1.67
Serum phosphate (mg/dL)	4.8 ± 1.3	Interdialytic weigh gain (Kg)	2.6 ± 0.8 kg

pro-BNP pro-brain natriuretic peptide, PTH: parathyroid hormone; nPCR: normalized protein catabolic rate.

### 3.3. Changes in Body Composition Over Time

Over two years, changes in post-dialysis body composition were analyzed in 124 dialysis patients. There were no significant changes in BMI (baseline vs. follow-up: 25.53 ± 4.92 vs. 25.63 ± 4.05, *p* = 0.077) but OH indicator significantly increased (−0.696 ± 1.67 vs. 0.268 ± 1.74; *p* = 0.007). The increase in OH indicator was paralleled by an increase in TBW (32.80 ± 6.85 vs. 35.91 ± 7.12; *p* < 0.001) at the expense of a significant increase in ECW (14.35 ± 2.66 vs. 16.01 ± 3.10; *p* <0.001), while ICW significantly decreased over time (19.90 ± 4.49 vs. 18.72 ± 4.12; *p* = 0.006) (Figure 1). There was a non-significant trend towards an increased ECW/ICW ratio (0.795 ± 0.129 vs. 0.850 ± 0.143; *p* = 0.055) over time but no significant changes in LTM or ATM.

### 3.4. Variables Associated with Low Post-Dialysis ICW

Baseline post-dialysis ICW was significantly positively correlated with serum creatinine, MUAMC, nPCR, 25 OH vitamin D, IDWG, LTM, and LTI; and negatively with age and CRP (Table 2). However, there was no significant correlation with pro BNP.

Additionally, baseline post-dialysis ICW was significantly lower in women than in men, in patients with malnutrition and in patients who had hypotension and dizziness during the dialysis session (Table 3).

The multivariate regression model with ICW as the dependent variable (Table 4) showed that male sex, MUAMC, and LTM were independently associated with higher ICW and inflammation, assessed as CRP; with lower ICW after adjustment for age, nPCR, plasma creatinine, and the presence of malnutrition (PEW).

### 3.5. Overhydration and Mortality: Low ICW as an Independent Risk Factor

Survival was determined after a median follow-up of 620 (IQR 248–961) days. In this period, 15 fatal events occurred, due to cardiovascular causes (*n* = 5), infection (*n* = 3), chronic disease deterioration (*n* = 4), neoplasia (*n* = 2), and unknown causes (*n* = 1). Baseline ICW levels of those who died were significantly lower than for survivors (15.09 ± 2.357 vs. 18.87 ± 4.525 L; *p* = 0.004). No other body fluid parameter displayed statistically significant differences, although there were non-significant trends to higher ECW/ICW ratio (0.779 ± 0.133 vs. 0.881 ± 0.100; P = 0.054), higher OH indicator (−0.85 ± 1.753 vs. 0.071 ± 1.186 L; *p* = 0.181), and lower TBW (33.08 ± 6.693 vs. 28.34 ± 4.108 L; *p* = 0.073) in patients who died (Figure 2). Despite the low number of events, Kaplan–Meier analysis showed that patients with lower ICW (*p*-value 0.018) and higher ECW/ICW ratio (*p*-value 0.021) were at increased risk of death, the cut-off values for maximum discrimination of all-cause mortality being ECW/ICW ratio 0.84 and ICW 17L (Figure 3).

In multivariable Cox proportional hazards models (Table 5), patients with lower ICW were at increased risk of dying even after adjustment for age, sex, comorbidity, CRP, and low albumin. However, the relationship between the ECW/ICW ratio and mortality was not significant.

## 4. Discussion

The present observational study identifies a decrease in ICW as a key component of subclinical overhydration. ICW was associated with muscle mass and male sex and negatively with inflammation. A high ECW/ICW ratio, a good fluid overload marker, was mainly driven by low ICW. In this regard, both low ICW and high ECW/ICW ratio were associated with mortality and patients with lower ICW values were at increased risk of dying even after adjustment.

Fluid overload is usually thought to consist of increased TBW at the expense of an increased ECW due to the loss of residual diuresis. Thus, there is a relationship between predialysis fluid overload and cardiovascular and all-cause mortality [1,2,3]. However, these studies were more focused on pre-dialysis BCM assessment to calculate ECW overload in order to adapt the dialysis prescription to achieve dry weight. Our study describes for the first time that despite an adequate ultrafiltration, changes in post-dialysis body composition over time consist of a significant increase in the OH parameter, despite a decrease in ICW and without a significant increase in ECW.

The significant ICW decrease over time is in line with other studies that have described the loss of lean tissue mass and its association with dialysis patient mortality [12,19]. We also observed that ICW was positively associated with indicators of muscle mass such as serum creatinine, MUAMC, and LTM [20], as well as a negative association between ICW and inflammation and malnutrition. Indeed, body cell mass depletion and abnormalities of the Na-K ATPase pump are observed in severely malnourished patients, eventually leading to decreased ICW and enhanced extravascular fluid shifts, resulting in ECW overload [21,22,23,24]. ICW was also associated with low energy intake, low interdialytic weight gain, and worse dialysis tolerance, with a higher prevalence of dizziness and hypotension. This could explain the controversial results of other studies, in which the highest fluid overload and the lowest interdialytic weight gain were associated with the highest mortality risk [25]. Patients with less IDWG would have less tolerance to ultrafiltration as fluid would be likely primarily stored in the interstitial and not in the intravascular space. Thus, cautious fluid removal strategy appears warranted.

The ECW/ICW ratio was a good marker of fluid overload in addition to being associated with mortality in our population [26]. In our study, the high ECW/ICW ratio was driven by the low ICW. In this regard, in other studies, a high ECW/ICW was associated with the presence of the MIA (malnutrition inflammation atherosclerosis) syndrome [7]. There are different formulas to express fluid overload such as the phase angle, ECW/TBW ratio, OH indicator (ECW/body weight), or the ECW/ICW ratio. In our study, both the ECW/ICW ratio and OH significantly increased over time, although only the ECW/ICW ratio was associated with mortality. Recently, the predialysis ECW/ICW ratio was shown to be a better marker of fluid overload than the ECW/TBW ratio and was also associated with all-cause mortality and CVD [26]. The OH indicator was developed by Chamney et al. [4], who proposed to calculate fluid overload as the difference between measured and the expected ECW which is expected in the sense of a normal renal function (ECW/body weight). We did not observe an association between OH indicator and mortality, likely because OH was assessed post dialysis and values were within the normal range. However, the ECW/ICW ratio increased due to a decreased ICW, despite normal OH indicator values. This does not detract from the value of predialysis OH, which is associated with mortality, to estimate ultrafiltration needs [27].

Finally, even with our reduced sample size, both low ICW and high ECW/ICW ratio were associated with mortality but only low ICW was independently associated with mortality in the multivariate Cox analysis. ICW is placed mainly in muscle mass, so the lower ICW represents mainly lower muscle mass, a well-known mortality predictive factor in dialysis patients and a criterion of the PEW syndrome [20]. Besides, low ICW is correlated with factors associated with PEW such as low energy intake and inflammation; all of which have been associated with mortality in previous studies [28,29]. In our study, patients who died presented significantly lower ICW and a non-significant trend towards higher ECW/ICW ratio. We postulate that solely low ICW better reflects the state of protein wasting and its association with mortality than the ECW/ICW ratio.

Certain limitations should be considered in the interpretation of the present findings. It was an observational study that precludes the assessment of causality. In addition, this study was performed at a single dialysis center and includes a small number of subjects and events. Therefore, results should be confirmed in larger prospective multicenter cohorts, ideally testing new approaches aimed at improving energy intake and avoiding muscle loss to try to avoid the trend towards decreasing ICW.

In conclusion, in hemodialysis patients, the post-dialysis OH parameter increased over time while ICW decreased, without changes in ECW. Low baseline post-dialysis ICW correlated with muscle wasting and inflammation and was an independent risk factor for mortality. This should encourage further research on the impact of implementing measures to increase energy intake or decrease protein catabolism and their impact on overhydration, ICW, and outcomes.

## Figures and Tables

**Figure 1 jcm-09-03616-f001:**
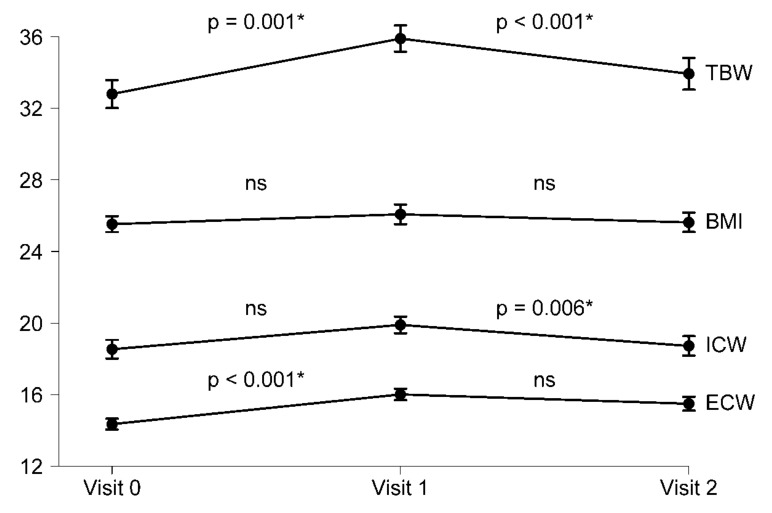
Progression of body composition in 124 hemodialysis patients over three years. TBW: total body water; BMI body mass index; ICW intracellular water; ECW extracellular water; ns no significance, * significance.

**Figure 2 jcm-09-03616-f002:**
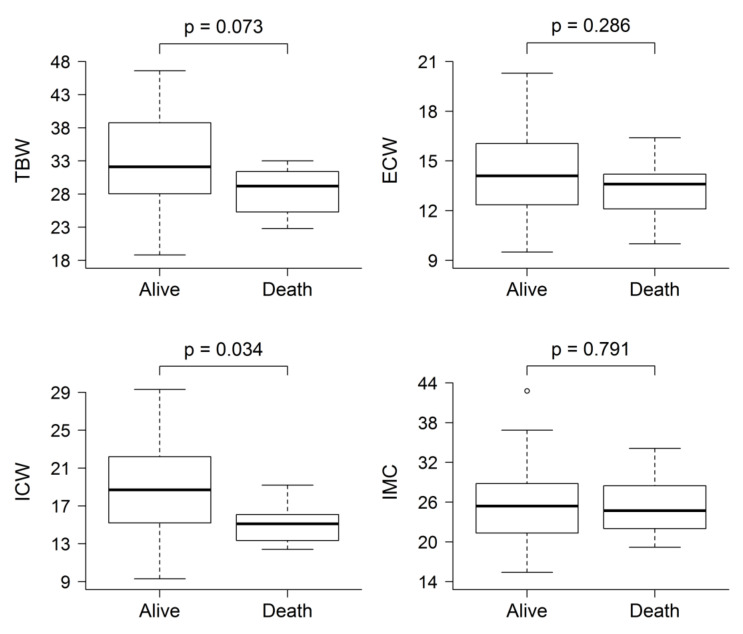
Comparison of baseline post-dialysis fluid status composition between patients who were dead and alive at last follow-up. The only body fluid parameter that was statistically significantly different was ICW.

**Figure 3 jcm-09-03616-f003:**
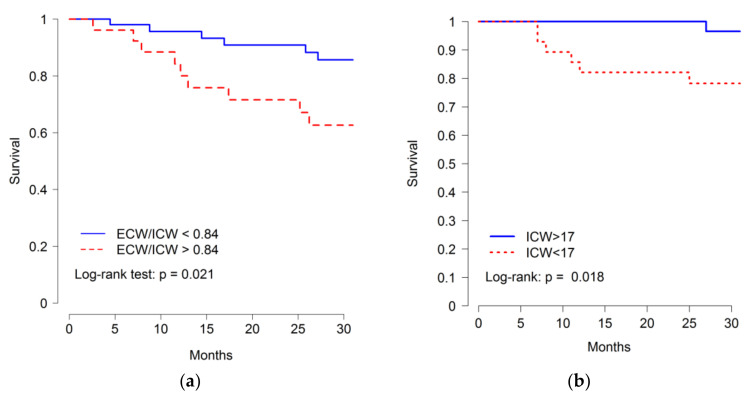
Kaplan–Meier curves for the time to death according to (**a**) ECW/ICW ratio and (**b**) ICW levels. Cut off for higher ECW/ICW ratio ≥0.84 and for lower ICW ≤ 17L.

**Table 2 jcm-09-03616-t002:** Univariate correlations of baseline post-dialysis ICF expressed as Spearman’s rank correlation coefficient.

Variable	r	*p*
Serum creatinine (mg/dL)	0.32	0.004
Mid upper arm muscle circumference (MUAMC)	0.41	<0.0001
The normalized protein catabolic rate (nPCR)	0.42	<0.0001
25 OH vitamin D	0.39	0.001
Interdialytic weight gain (IDWG)	0.46	<0.0001
Lean tissue mass (LTM)	0.56	<0.0001
Lean tissue index (LTI)	0.86	<0.0001
Age	−0.30	0.007
Serum C Reactive Protein (CRP)	−0.28	0.014
pro-brain natriuretic peptide (pro BNP)	−0.22	0.051

**Table 3 jcm-09-03616-t003:** Baseline post-dialysis ICF (L) in subgroups of patients.

Variable	Subgroups		*p*-Value
Sex	Women (*n* = 33)	Men (*n* = 44)	<0.0001
Baseline post-dialysis ICF (L)	15.6 ± 3.0	20.8 ± 4.2	
Malnutrition (presence PEW)	Yes (*n* = 31)	No (*n* = 45)	0.021
Baseline post-dialysis ICF (L)	17.1 ± 4.7	19.6 ± 4.1	
Hypotension during dialysis	Yes (*n* = 24)	No (*n* = 51)	<0.0001
Baseline post-dialysis ICF (L)	15.7 ± 3.7	19.8 ± 4.4	
Dizziness during dialysis	Yes (*n* = 22)	No (*n* = 53)	0.02
Baseline post-dialysis ICF (L)	16.6 ± 4.3	19.3 ± 4.5	

**Table 4 jcm-09-03616-t004:** Multivariable linear regression analyses to test de independent relationship between ICW and other clinical variables.

Model/Variables	β Coefficient	SE of the Estimate	95% CI	*p*-Value
Model 1 (R^2^ = 0.834)				
Lean tissue mass	0.162	0.014	0.132–0.189	<0.0001
Mid upper arm muscle circumference	0.74	0.067	0.606–0.875	<0.0001
Sex	3.001	0.465	2.072–3.930	<0.0001
Model 2 (R^2^ = 0.845)				
Lean tissue mass	0.152	0.014	0.123–0.181	<0.0001
Mid upper arm muscle circumference	0.726	0.066	0.594–0.857	<0.0001
Sex	3.131	0.458	2.215–4.046	<0.0001
C-Reactive Protein	−0.235	0.115	(−0.465)–(−0.004)	0.046

Model 2 suggests an independent association between C–reactive protein and ICW after the significant influence of some confounders.

**Table 5 jcm-09-03616-t005:** Adjusted relative risk of all-cause mortality in 122 prevalent hemodialysis patients.

Variable	HR	*p*
Age	1.197 (1.051–1.364)	0.007
Male	0.977 (0.119–8.044)	0.983
Davies medium	11.4 (0.419–312.8)	0.149
Davies high	1.213 (0.060–24.58)	0.900
ICW	0.627 (0.401–0.979)	0.040
C-Reactive Protein	1.150(0.803–1.647)	0.446
Albumin ≤ 3.8 g/dL	0.037(0.003–0.438)	0.009

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
