# Peer review of "Low Intracellular Water, Overhydration, and Mortality in Hemodialysis Patients"

_jcm, 2020, doi:10.3390/jcm9113616_

Round 1
Reviewer 1 Report
I find the article interesting however some changes shoud be made before the publication.
Suggestions:
- Measurements in Body Composition Monitor FMC are based on bioimpedance spectroscopy so please use BIS rather than BIA
- The body compartments in BCM FMC are named as Extracellular Water ECW, and ICW , TBW please use uniform nomenclature
- Spearman rank correlation coefficient in the most papers is marked as r (it is more popular)
- Results: AMC - please explain the abreviation
- Line 171: change the order of values (15.09 L is lower than 18.87 L)
- Discussion: Intracellular water is placed mainly in muscle mass so it obvious that it correlates with LTM and LTI. Patients who died during the follow-up had significantly lower ICW because it represents amount of muscle mass. Muscle loss is well known mortality predictive factor in patients with chronic kidney disease. Please emphasize it in the discussion rather than be concentrated on the fluid disturbances. You did the BIS measurements after dialysis session so it was the proper moment for body composition measurements.
Author Response
Reviewer 1
- Measurements in Body Composition Monitor FMC are based on bioimpedance spectroscopy so please use BIS rather than BIA
- R: revised as requested, we have changed to BIS instead BIA
- The body compartments in BCM FMC are named as Extracellular Water ECW, and ICW , TBW please use uniform nomenclature
- R: revised as requested, the nomenclature TBF, ECF, ICF have changed to TBW, ECW and ICW
- Spearman rank correlation coefficient in the most papers is marked as r (it is more popular)
- R: revised as requested
- Results: AMC - please explain the abreviation
- R: We apologize, thanks for pointing this out. It should be MUAMC mid-upper arm muscle circumference
- Line 171: change the order of values (15.09 L is lower than 18.87 L)
- R: revised as requested, thanks for pointing this out.. (line 449)
- Discussion: Intracellular water is placed mainly in muscle mass so it obvious that it correlates with LTM and LTI. Patients who died during the follow-up had significantly lower ICW because it represents amount of muscle mass. Muscle loss is well known mortality predictive factor in patients with chronic kidney disease. Please emphasize it in the discussion rather than be concentrated on the fluid disturbances. You did the BIS measurements after dialysis session so it was the proper moment for body composition measurements.
- R: As suggested, the discussion about low intracellular water and increased risk of mortality now emphasizes this. Line 577-579:“ICW is placed mainly in muscle mass, so the lower ICW represents mainly lower muscle mass, well known mortality predictive factor in dialysis patients and criteria of the PEW syndrome (20)”

Reviewer 2 Report
Summary
This is a prospective study analyzing post-dialysis changes in ICF/ECF as measured by BIA with nutrition and inflammatory markers and mortality in 124 MHD patients.
Major issues
The introduction needs to be expanded, especially an explanation about the BIA (ECF/ICF); why the BIA the authors chose is appropriate for this study.
Lines 60-62: The original published study is written in Spanish. The authors should briefly describe the original study design. Is this a secondary analysis or a post hoc analysis of the same study design?
Line 92: More detail should be provided about the BIA – single or multiple frequency?
Line 103-104, if IRSNM criteria was used to identify PEW, how did the authors justify using PCR as low protein intake – is there a reference for using PCR for PEW criteria?
Results section: patient characteristics should be reported in a table as well as results from the BIA
Line 137: Explanation for PEW determination confusing (were all three labs: Alb, PAB, and cholesterol, measured for all subjects and used to determine PEW?) There are 4 PEW categories, and criteria within those categories. Subjects must criteria within 3 out of 4 categories to be classified as having PEW.
143 -144 Data should be presented in a table
Figure 1: What type of statistical analysis was used to analyze changes over time?
158-166 Should be presented in a table; should not use “significant trend” if alpha = 0.05, Check p values; consistency issues with data presentation; missing symbols.
192-193: Associations with ICF and muscle, inflammation – were these correlations?
Author Response
Reviewer 2
-The introduction needs to be expanded, especially an explanation about the BIA (ECF/ICF); why the BIA the authors chose is appropriate for this study.
R: Thanks for the suggestion, we have clarified in the introduction why the use of BIA and specially multifrequency-BIA is appropriate for studies in dialysis. However, following reviewer 1, BIA was replaced by BIS:
Line 80- 83 “The recent KDOQI clinical practice guideline for nutrition in CKD 2020 suggest the use of multifrequency BIS (MF-BIS) to assess body composition. FM (fat mass) and FFM (fat free mass) measured using MF-BIS had good agreement with DXA and had high correlations with several markers of nutritional status”
- Lines 60-62: The original published study is written in Spanish. The authors should briefly describe the original study design. Is this a secondary analysis or a post hoc analysis of the same study design?
R: We have now described briefly in the patients and methods section the original study design, Line 96-107: Following the reviewer question, we believe the reviewed suggestion that this s is a secondary analysis is correct. Additional info has been added as follows:
“This is a secondary analysis of a prior cross-sectional study with prospective follow-up of prevalent dialysis patients conducted at the Iñigo Alvarez de Toledo Kidney Foundation (FRIAT) hemodialysis center of the Fundación Jiménez Díaz Hospital, Madrid, Spain (12). All prevalent patients (n=124) on maintenance hemodialysis at the dialysis center were screened and the study ran from January 1, 2010 and survival was calculated until October 1, 2012 with a median of follow-up of 461 days (240-931 days). The study consisted of three visits in which clinical, biochemical, anthropometric and body composition parameters were collected along with dialysis characteristics and the presence of Protein Energy Wasting (PEW) was evaluated annually (baseline, 12 and 24 months).”
Although the original article was published in the journal Nefrología and can be accessed in both Spanish and English, we could not directly access it through pubmed as the link appears to have become corrupted and instead had to google it. Thus, we thank the reviewer for the suggestion that improves the reader experience.
- Line 92: More detail should be provided about the BIA – single or multiple frequency?
R: “To assess body composition we used the Fresenius Medical Care BCM body composition monitor that is a multi-frequency Bio-impedance spectroscopy device. We have emphasized that is multi-frequency in the text. Line 191-192
- Line 103-104, if IRSNM criteria was used to identify PEW, how did the authors justify using PCR as low protein intake – is there a reference for using PCR for PEW criteria?
R: The referee is correct. There is indeed a reference. The normalized protein catabolic rate (nPCR), expressed as g/kg per day is a marker of dietary protein intake in dialysis patients (reference: Jindal KK, Goldstein MB. Urea kinetic modeling in chronic hemodialysis: Benefits, problems, and practical solutions. Semin Dial. 1988; 1:82-85) . nPCR = (0.0136 x [Kt/Vurea × ([predialysis BUN + postdialysis BUN] ÷ 2)]) + 0.251. Indeed, according PEW criteria, an unintentional low dietary protein intake <0.80 g kg-1 day-1 for at least 2 months is a criteria of PEW. The original description of PEW states that: “Can be assessed by dietary diaries and interviews, or for protein intake by calculation of normalized protein equivalent of total nitrogen appearance (nPNA or nPCR) as determined by urea kinetic measurements” (Fouque D, Kalantar-Zadeh K, Kopple J, et al. A proposed nomenclature and diagnostic criteria for protein-energy wasting in acute and chronic kidney disease. Kidney Int. 2008;73(4):391-398.)
- Results section: patient characteristics should be reported in a table as well as results from the BIA
R: Done it, line 280 and 288, in table 1 we describe the baseline characteristics of the study population
Table1.- Baseline characteristics of the 124 dialysis patients. Values expressed as mean±SD, % (n) or median (IQR).
Variable |
Value |
Variable |
Value |
Age (years) |
64±13 |
Serum C Reactive Protein (mg/dl) |
1.3±0.6 |
Women, n (%) |
45 (49) |
Serum albumin (g/dL) |
3.8±0.4 |
Diabetes, n (%) |
21 (23) |
nPCR (g/kg/day) |
0.7±0.3 |
Dialysis vintage (months) |
59±79 |
Plasma BNP (pg/ml) |
6090 (2407-14900) |
Cardiovascular disease, n (%) |
40 (44) |
Plasma total CO2 (mmol/L) |
20±3 |
Predialytic systolic blood pressure (mmHg) |
132±24 |
Total Body Water (L) |
32.8 ± 6.8 |
Predialytic diastolic blood pressure (mmHg) |
67±14 |
Extracellular Water (ECW) (L) |
14.4 ± 2.6 |
Protein Energy Wasting, n (%) |
44 (41%) |
Intracellular Water (ICW) (L) |
18.5 ± 4.5 |
Kt/Vurea |
1.42±0.18 |
ECW/ICW |
0.79 ± 0.13 |
Hemoglobin (g/dl) |
11.6±1.4 |
Lean Tissue Mass (kg) |
58±16 |
Plasma iPTH (pg/mL) |
250 (127-454) |
Adipose Tissue Mass (kg) |
30±11 |
Serum calcium (mg/dl) |
8.8±0.8 |
Overhydration (L) |
-0.70 ± 1.67 |
Serum phosphate (mg/dl) |
4.8±1.3 |
Interdialytic weigh gain (Kg) |
2.6±0.8 kg |
pro-BNP pro-brain natriuretic peptide, PTH: parathyroid hormone; nPCR: normalized protein catabolic rate.
- Line 137: Explanation for PEW determination confusing (were all three labs: Alb, PAB, and cholesterol, measured for all subjects and used to determine PEW?) There are 4 PEW categories, and criteria within those categories. Subjects must criteria within 3 out of 4 categories to be classified as having PEW.
R: As requested, further clarification was added to the text lines 204-213:” The diagnosis of malnutrition, as defined by PEW, requires the joint evaluation of the combination of clinical, anthropometric and body composition parameters using bioelectrical impedance (BIA multifrequency) and biochemical parameters. PEW was assigned to patients meeting at least three criteria in the four different categories for malnutrition markers at baseline: biochemical (albumin< 3.8 g/dl, prealbumin < 30 g/dl or cholesterol < 100 mg/dl); body mass (BMI< 23 kg/m2, weight loss >5% in 3 months or >10% in 6 months or body fat <10%); muscle (muscle mass loss >5% in 3 months or >10% in 6 months or MUAMC reduction ≥10% associated with the 50th percentile in the population); Protein intake (nPCR <0.8 g/kg/day)”
- 143 -144 Data should be presented in a table
R: Already described in table 1
Figure 1: What type of statistical analysis was used to analyze changes over time?
R: Line 256-257. This has been clarified as follows: “Comparison of the same variable at different visits was carried out using the Wilcoxon test for related samples”
158-166 Should be presented in a table; should not use “significant trend” if alpha = 0.05, Check p values; consistency issues with data presentation; missing symbols.
R: A table has been added summarizing these data and “significant trend” was deleted. The paragraph now reads as follows:
“Baseline post-dialysis ICW was significant positively correlated with serum creatinine, MUAMC, nPCR, 25 OH vitamin D, IDWG, LTM and LTI; and negatively with age and CRP (Table 2). However, there was no significant correlation with pro BNP.Line 353-358
Additionally, baseline post-dialysis ICW was significantly lower in women than in men, in patients with malnutrition and in patients who had hypotension and dizziness during the dialysis session. (Table 3).” line 438
Table 2.-Univariate Correlations of baseline post-dialysis ICW expressed as Spearman´s rank correlation coefficient
Variable |
R |
p |
Serum creatinine |
0.32 |
0.004 |
Mid upper arm muscle circumference (MUAMC) |
0.41 |
<0.0001 |
The normalized protein catabolic rate (nPCR) |
0.42 |
<0.0001 |
25 OH vitamin D |
0.39 |
0.001 |
Interdialytic weight gain (IDWG) |
0.46 |
<0.0001 |
Lean tissue mass (LTM) |
0.56 |
<0.0001 |
Lean tissue index (LTI) |
0.86 |
<0.0001 |
Age |
-0.30 |
0.007 |
Serum C Reactive Protein (CRP) |
-0.28 |
0.014 |
pro-brain natriuretic peptide (pro BNP) |
-0.22 |
0.051 |
Table 3. Baseline post-dialysis ICW (L) in subgroups of patients
Variable |
Subgroups |
P value |
|
Sex |
Women (n=33) 15.6 ± 3.0 |
Men (n=44) 20.8 ± 4.2 |
<0.0001 |
Malnutrition (presence PEW) |
Yes (n=31) 17.1±4.7 |
No (n=45) 19.6±4.1 |
0.021 |
Hypotension during dialysis |
Yes (n=24) 15.7 ±3.7 |
No (n=51) 19.8± 4.4 |
<0.0001 |
dizziness during dialysis |
Yes (n=22) 16.6 ±4.3 |
No (n=53) 19.3± 4.5 |
0.020 |
192-193: Associations with ICF and muscle, inflammation – were these correlations?
R: As suggested by another reviewer, we have performed a multivariate regression analysis between ICW and the variables that were significant in the univariate analysis, both C-Reactive Protein (indicator of inflammation) and Mid Upper Arm Muscle Circumference (indicator of muscle mass) were independently associated with ICW (line 359-362 ) table 4 (line 440)
Table 4.- Multivariable linear regression analyses to test the independent relationship between ICW and other clinical variables. MUAMC: Mid upper arm muscle circumference
Model/variables |
β coefficient |
SE of the estimate |
95% CI |
pvalue |
Model 1 (R2=0.834) Lean tissue mass MUAMC Sex |
0.162 0.740 3.001 |
0.014 0.067 0.465 |
0.132-0.189 0.606-0.875 2.072-3.930 |
<0.0001 <0.0001 <0.0001 |
Model 2 (R2=0.845) Lean tissue mass MUAMC Sex C-Reactive Protein |
0.152 0.726 3.131 -0.235 |
0.014 0.066 0.458 0.115 |
0.123-0.181 0.594-0.857 2.215-4.046 (-0.465)- (-0.004) |
<0.0001 <0.0001 <0.0001 0.046 |
Model 2 suggests an independent association between C - reactive protein and ICW

Reviewer 3 Report
The study aims to analyse the impact of excess fluid removal during dialysis on post dialysis ICF and ECF and its hemodynamic tolerance. The description of the aim of the study is too general and an effort to make it more clear is due.
The study consists of a post-hoc analysis, it starts from a cohort got back from a cross-sectional study and followed-up for about two years. The primary outcome is not clearly defined and, consequently, there is no formal definition of the sample size. The Authors should review this point.
In the "Patients and Methods" section the description of the variables used in the analysis is accurate, also the description of the statistical methodology is accurate, but it is affected by an unclear definition of the outcomes.
In the "results" section, after the report of the patients characteristics and the baseline post-dialysis body composition, the trends of OH, TBF (in the legend of figure 1 TBF is substituted by TBW), BMI, ICF and ECF are described. Since the most relevant result is the decrease of ICF, the Authors look for the associations of ICF with other variables, and many of them show a significant association, thus I suggest also a multivariate regression. At last, the survival analysis shows that only age and ICF are predictors of death.
The findings of the study lead to the conclusion that a reduction of ICF determines unfavourable outcomes, the role of the "subclinical" OH is only a hypothesis, not truly supported by data. This point should be considered in the discussion.
Author Response
Reviewer 3
Comments and Suggestions for Authors
-The study aims to analyse the impact of excess fluid removal during dialysis on post dialysis ICF and ECF and its hemodynamic tolerance. The description of the aim of the study is too general and an effort to make it more clear is due.
R:Thank you for the comment, we have clarified the aim as follows in the abstract and introduction as follows (line 98-102)
“The primary aim of the present study was to analyze changes in post-dialysis ECW, intracellular water (ICW) and the overhydration (OH) parameter over time. Additionally, the association of these variables with mortality was explored.”
-The study consists of a post-hoc analysis, it starts from a cohort got back from a cross-sectional study and followed-up for about two years. The primary outcome is not clearly defined and, consequently, there is no formal definition of the sample size. The Authors should review this point.
In the "Patients and Methods" section the description of the variables used in the analysis is accurate, also the description of the statistical methodology is accurate, but it is affected by an unclear definition of the outcomes.
R: We have now revised this point as follows line 105-107 and line 256-263:
“The primary aim was to analyze changes in post-dialysis (i.e. following removal of excess ECW) ECW, intracellular water (ICW) and the overhydration (OH) parameter over time. For this, no formal sample size was calculated. Next, we explored the variables associated with these changes. Finally, we explored the association of these variables with mortality”
-In the "results" section, after the report of the patients characteristics and the baseline post-dialysis body composition, the trends of OH, TBF (in the legend of figure 1 TBF is substituted by TBW), BMI, ICF and ECF are described. Since the most relevant result is the decrease of ICF, the Authors look for the associations of ICF with other variables, and many of them show a significant association, thus I suggest also a multivariate regression. At last, the survival analysis shows that only age and ICF are predictors of death.
R: We thank the reviewer for this suggestion. We have now performed the suggested analysis and incorporated it into the manuscript, line 359-362. Multivariate regression modelling with ICW as the dependent variable (Table 4) showed that CRP, sex. MUAMC and LTM were independently associated to ICW after adjustment for age, nPCR, plasma creatinine, and the presence of malnutrition (Protein Energy Wasting).
Table 4.- Multivariable linear regression analyses to test the independent relationship between ICW and other clinical variables. MUAMC: Mid upper arm muscle circumference (line 440)
Model/variables |
β coefficient |
SE of the estimate |
95% CI |
pvalue |
Model 1 (R2=0.834) Lean tissue mass MUAMC Male sex |
0.162 0.740 3.001 |
0.014 0.067 0.465 |
0.132-0.189 0.606-0.875 2.072-3.930 |
<0.0001 <0.0001 <0.0001 |
Model 2 (R2=0.845) Lean tissue mass MUAMC Male sex C-Reactive Protein |
0.152 0.726 3.131 -0.235 |
0.014 0.066 0.458 0.115 |
0.123-0.181 0.594-0.857 2.215-4.046 (-0.465)- (-0.004) |
<0.0001 <0.0001 <0.0001 0.046 |
Model 2 suggests an independent association between C - reactive protein and ICW
The findings of the study lead to the conclusion that a reduction of ICF determines unfavourable outcomes, the role of the "subclinical" OH is only a hypothesis, not truly supported by data. This point should be considered in the discussion.
R: As suggested, we have changed the conclusion with respect to OH as follows: “changes in post-dialysis body composition over time consist of a significant increase in the OH parameter, despite a decrease in ICW and without a significant increase in ECW.” line 590-592.

Round 2
Reviewer 2 Report
Noted revisions accepted
Reviewer 3 Report
The Authors have substantially answered the questions posed by the reviewer.